# Forage:Concentrate Ratio Effects on In Vivo Digestibility and In Vitro Degradability of Horse’s Diet

**DOI:** 10.3390/ani13162589

**Published:** 2023-08-11

**Authors:** Fabio Zicarelli, Raffaella Tudisco, Daria Lotito, Nadia Musco, Piera Iommelli, Maria Ferrara, Serena Calabrò, Federico Infascelli, Pietro Lombardi

**Affiliations:** Department of Veterinary Medicine and Animal Production, University of Naples Federico II, 80137 Napoli, Italy; fabiozicarelli@gmail.com (F.Z.); piera.iommelli@unina.it (P.I.); maferrar@unina.it (M.F.); federico.infascelli@unina.it (F.I.); pietro.lombardi@unina.it (P.L.)

**Keywords:** horses, digestibility, degradability, in vitro gas production

## Abstract

**Simple Summary:**

Contrary to other herbivores, in horses the forage:concentrate ratio of the diet may be critical according to the animal attitude and workload. In this study, diet digestibility and degradability (in vivo and in vitro, respectively) were assessed in five horses’ diets that differed in the percentage of concentrates that, in adult horses, are considered more as a hay supplement rather than a basic feed. The determination of in vivo digestibility and in vitro degradability to assess the nutritional characteristics of the horses’ diet shows how this approach can be of useful to determine concentrate intake and optimize the energy content in a horse’s diet.

**Abstract:**

Determination of digestibility represents the first step for the evaluation of the net energy content of feed for livestock animals. The aim of this study was to evaluate the in vivo digestibility and in vitro degradability of five diets characterized by different forage/concentrate ratios (F:C) in horses. The in vitro degradability was determined by the Gas Production Technique (GPT), using as an inoculum source the feces of the same subjects used for the in vivo test. Five diets consisting of poliphyte hay, straw and grains of barley and oats with a different F:C ratio [90/10 (Diet 1); 78/22 (Diet 2); 68/32 (Diet 3); 60/40 (Diet 4); 50/50 (Diet 5) were formulated and administered in succession, starting with Diet 1. In the in vivo results, no significant differences emerged, despite the different F:C content. In in vitro fermentation, four diets out of the five (2, 3, 4, 5) presented a similar trend of the curve of gas production, showing good activity of the fecal micro population during the first hours of incubation. An important correlation between gas and Volatile Fatty Acid (VFA) were found, suggesting that the processes linked to the micro population deriving from the horse’s caecum follow metabolic pathways whose products can be modeled in the same way as for the rumen. The GPT could represent the correct method for studying the nutritional characteristics of feed for horses, using feces as the source of inoculum, even if further investigations must be performed to improve the technique.

## 1. Introduction

The determination of digestibility represents the first step for the evaluation of feed net energy; in horses, digestibility can be determined in vivo by the ingesta–excreta balance or by the marker method [1]. In addition, feed digestibility can be estimated by chemical composition parameters [2], by NIRS method [3,4] by the “in sacco” method [5], and by the in vitro pepsin-cellulase technique [6]. These in vitro techniques arouse particular interest because they are not expensive and are easy to perform. Among these, the cumulative gas production technique (GPT) allows us to study the fermentation kinetics and the digestibility of the organic matter in feed. GPT is based on the anaerobic degradation of carbohydrates by the micro-population of the digestive tract with the production of volatile fatty acids [7,8,9], carbon dioxide, and methane [10].

Menke and Steingass [10] found a close correlation between gas production measured after 24 h of incubation using rumen fluid as inoculum, and in vivo digestibility. Khazaal et al. [11] reported that, in sheep, the relationship between in vivo dry matter digestibility and volume of gas produced was very close (r = from 0.84 to 0.81; *p* < 0.01) after 3 and 6 h, respectively. In horses, the GPT is performed using feces as the source of inoculum [12].

Diet digestibility is influenced by several factors, one the most important being the forage/concentrate ratio [13,14], which has also been reported to affect animal health [15], feeding behavior [16] and healthy characteristics of foods of animal origin [17,18,19,20,21].

In horses, the forage:concentrate ratio is strictly linked with the animals’ attitude and workload. Forage is the most important feed in a horse’s diet, often providing most of the nutrients fed to horses [21]. Administering the right quantity and type of forage is critical, and hay can also give rise to health problems, depending on the quality and quantity of its components. Health issues can arise either when the level of forage fed is inadequate, the quality of forage is not good enough, or the digestibility is not appropriate for the life-stage or activity of the horses [22]. Forage contributes to the overall energy and nutrient content of a horse’s ration, but also helps to maintain digestive health through its physical effect on the movement of food through the gut, as well as through the retention of fluid within the digestive tract.

In horses, forage should not be seen as a ‘filler feed’, or just something to keep a horse occupied between hard feeds, as it makes a very positive contribution to the overall ration. Making good choices with regards to forage will help to maintain digestive function. In order to provide an energy source for horses, rations often include starch rather than fiber. This can result in health issues related to the gastrointestinal tract (GIT) in the horse [23]. In fact, forage contributes to the overall energy and nutrient content of a horse’s ration, but also helps to maintain digestive health through its physical effect on the movement of feed through the gut, as well as through the retention of fluid within the digestive tract.

By contrast, concentrates in the horse diet should only be considered as a good-quality hay supplement. In general, a mature horse does not require the energy that would be provided by concentrated feeds (cereals/sweet feeds, pellet feeds, etc.) unless the horse is used for more than light work and/or production, such as a nursing mare or a breeding stallion [24]. Horses are more frequently overfed rather than underfed, and this is often due to an excess or an improper use of concentrates in the diet [25]. Concentrates, however, play an important part of the growing foal’s diet through maturity, even contributing up to 50% of the ration in the first 2–3 years of growth. Thereafter, unless there are high energy and/or growth needs given current age and work level, slight increases in hay can provide the extra energy to balance dietary needs.

In order to make a contribution to this topic, this trial has been performed to study the correlations between the in vivo digestibility and in vitro degradability of five diets with different forage/concentrate ratios (F:C) in horses. The in vitro degradability was determined with GPT, using as an inoculum source the feces of the same subjects used for the in vivo test [26].

## 2. Materials and Methods

### 2.1. Animals and Diets

Four six-year-old horses with a live weight of 500 ± 22 kg were included in the trial. Animals were kept in individual stalls to facilitate the control of feed intake and feces collection.

Five diets consisting of polyphyte hay, straw and grains of barley and oats with a different forage:concentrate ratios (F:C) of 90/10 (Diet 1); 78/22 (Diet 2); 68/32 (Diet 3); 60/40 (Diet 4); and 50/50 (Diet 5) were formulated and administered in succession from the highest to the lowest amount of forage. Diet 1 was the one administered to animals before the onset of the experiment.

### 2.2. Chemical Composition

The chemical composition was determined on the obtained samples according to the protocol suggested by AOAC [26]. In particular, the ingredients of the diets were ground through a 1 mm grid with a mill (Brabender Wiley mill, Braebender OHG, Duisburg, Germany) and mixed in the same proportion present in the diets. Feces were ground with the same technique, and the organic substance content was determined [27].

Acid-insoluble ash in diets and feces was determined by the method of Bergero et al. [28]. Such a method allows one to determine the content of mineral substances insoluble in hydrochloric acid. Briefly, the sample is deposited in a 500 mL flask to which 100 mL of 4N hydrochloric acid are added. The flask is then brought to a boil for 30 min. The hot solution is filtered (Wathman filters No. 41), and the residue is washed with hot water until the acid reaction disappears. Subsequently, the filter is transferred into a pre-weighed porcelain capsule which, after drying, is placed in a muffle at 650 °C for the determination of the ashes, which are related to the quantity of weighed dry substance.

### 2.3. In Vivo Digestibility

An adaptation period of 14 days was foreseen for each diet. During this time, individual voluntary intake was evaluated in two daily meals (at 8:00 and 16:00). Therefore, a 6-day trial period started, during which each animal received 90% of the amount of dry matter previously ingested to avoid residues. Individual stool sampling (about 200 g) was performed directly from the rectum three times a day (always at the same time to reduce the effect of the variability of their composition throughout the day). The individual daily pool of feces was homogenized, an aliquot was used to determine the dry matter content at 103 °C, and another one was dried at 65 °C and used to prepare the individual pool of six test days for each horse. Similarly, for each diet, a sample was created daily to be associated with the feces pool of each animal for the determination of digestibility. The digestibility of the organic substance was evaluated with the internal indicator method, using the insoluble acid ash using the following formula:ADC = [(Cf − Ca) / Cf] × 100(1)
where ADC is the apparent digestibility coefficient of organic matter and crude fiber, and Cf and Ca represent the concentration of the AIA with respect to organic matter content in feces and diet, respectively.

### 2.4. In Vitro Degradability

On the last day of the in vivo tests, a feces sample was taken from each animal, kept in anaerobic conditions at a temperature of 39 °C, immediately transported to the Food analysis laboratory of the Department of Veterinary Medicine and Animal Production, and used for the preparation of the inoculum for the in vitro test, using the GPT [29,30]. To this end, according to Macheboeuf et al. [31], 50 g of feces was mixed with 100 mL of anaerobic buffer at 39 °C, filtered through four layers of gauze, and diluted 1:1 with the buffer, finally obtaining an inoculum for each of the four horses used for each diet. For each diet, about 1.0 g of sample was placed in a 120 mL serum bottle which, after adding 75 mL of medium and 4 mL of reducing agent, was hermetically sealed with a butyl rubber stopper and aluminum, and placed in a thermostat at 39 °C until the internal temperature was balanced. All the steps were carried out under CO_2_ insufflation to maintain the anaerobiosis. Thereafter, the bottles were added with 10 mL of inoculum and, after having balanced their internal pressure with the atmospheric one, they were incubated in a thermostat at 39 °C.

For each inoculum, coming from a single animal, 3 replications were carried out to have an average value of all GPT parameters. Furthermore, two bottles were incubated without feed and used as blank. At pre-established times, with intervals of 2–24 h, 20 gas measurements were taken for each bottle using a manual system consisting of a pressure transducer (Cole and Palmer Instrument Co., Vernon Hills, IL, USA) inserting a 21 G × 1″ (0.80 × 25 mm) needle through the vial caps that were attached to the pressure transducer. Then, the transducer was removed, and the needle was inserted into the cap for a few seconds for complete stabilization between internal and external pressures. The gas pressure (psi) measured during the test was transformed into volume (mL of gas). At the end of the gas readings, the bottles were shaken to mix the suspension.

After 120 h of incubation, the bottles were opened, and the pH was determined using a pH meter (ThermoOrion 720 A+, Fort Collins, CO, USA). Subsequently, an aliquot of the liquid present in the bottle was taken to determine the volatile fatty acids (VFA) by gas chromatography (ThermoQuest mod. 8000top, FUSED SILICA capillary column 30 m × 0.25 mm × 0.25 mm film thickness) according to Formato et al. [32]. After that, the content of each bottle was filtered through pre-weighed porous septum crucibles (Schott-Duran #2), which were placed in an oven at 103 °C and then in a muffle at 550 °C to estimate the residual organic matter; the degraded organic matter (dMO) was calculated by the difference between the incubated one and the residual one, corrected for the blank. The total gas production (corrected for the blank) was related to the incubated organic matter (OMCV, mL/g) and the degraded organic substance (Yield, mL/g).

### 2.5. Statistical Analyses

For each bottle, the cumulative volumes of gas obtained were related to the incubated organic matter and processed with the one-phase Michaelis–Menten model modified by Groot et al. (1996) [33]:(2)G=A/(1+BtC)
where G represents the quantity of gas (mL/d) produced at time t; A is the potential gas production (mL/g); B is the time (h) necessary to produce a quantity of gas equal to A/2; and C is a constant that defines the shape of the curve.

All data relating to in vivo digestibility and GPT parameters were processed by ANOVA using the General Linear Model (GLM) procedure, including the group effect as a fixed effect and the month of sampling as a repeated measure. The differences between means were evaluated with the T-test.

Furthermore, to evaluate the relationships between in vivo and in vitro results, the correlation and possible regressions (CORR and REG procedures, respectively, of SAS, 2000) between the digestibility coefficients and the GPT parameters (dOM, OMCV, A, B, Yield, VFA) were determined.

## 3. Results

Table 1 shows the ingredients and the chemical composition of the five diets. Diets 1 and 2 showed the highest crude fiber and the lowest crude protein values (30 and 30.7% DM, 6.4 and 6.3% DM, respectively). The lowest content of CF (21.2% on DM basis) and the highest content of CP (8.7% DM) was found for Diet 4. In any case, the chemical composition was congruous with the contribution of the various ingredients (polyphite hay, straw, grains).

### 3.1. In Vivo Results

Dry matter ingestion varied between 10.7 and 11.2 kg/d (Table 2), and no significant differences emerged among the diets, despite the differences in F:C ratio. Table 3 shows the apparent digestibility coefficients of organic matter (OM) and crude fiber (CF).

Diet 2, characterized by a higher intake of straw, showed the lowest apparent digestibility coefficient (ADC) for organic matter, significantly different (*p* < 0.01) than the other four. This result is in line with the CF content for Diets 3, 4, 5, while for Diet 1, which has a CF content similar to Diet 2, the higher digestibility observed for OM is attributable to the absence of straw. As far as crude fiber is concerned, the lowest values were obtained for Diets 4 and 5.

### 3.2. In Vitro Results

The in vitro fermentation parameters are shown in Table 4. The Groot model has always proved to be suitable to describe the cumulative gas production profile; in fact, the R2 values are between 0.975 and 0.998. Diet 3 showed the highest (*p* < 0.01) potential gas production (A = 313 mL/g) and the lowest time needed to produce a quantity of gas equal to A/2 (B = 22.1 h), indicating a faster fermentation process. The same diet also presented the greatest OM degradability (66%) associated with the highest real gas production (OMCV: 274 mL/g). The lowest dOM value (54.99%), significantly different (*p* < 0.01) from that obtained for Diets 3 and 4, was found in Diet 2. The pH was significantly highest (*p* < 0.01) in Diet 3 and significantly lowest in Diet 4; intermediate values were registered for the other diets. If Diet 4 is excluded, the pH at the end of the incubation remained for all the diets at values compatible with the full efficiency of the cellulolytic bacteria (pH = 6.4), as reported by van den Berg et al. [35].

Figure 1 shows the trend of the cumulative gas production estimated by the model as a function of time. Diet 3 recorded the greatest gas production at each time since the first hours of incubation, whereas Diet 1, unlike all the others, is associated with a slight lag phase, with the lower gas production along the most incubation time.

Diet 3 (Table 5) showed significantly (*p* < 0.01) higher values of total VFA and acetate, while in Diet 5, the values were lower.

The study of the regressions of the gas production on that of VFA and of the in vivo digestibility of OM on the ADC parameters has highlighted some noteworthy results. For each mmol of VFA produced, 1.69 mmols of gas was obtained (Table 6, a value very similar to the theoretical value (1.41) expected from Wolin’s balance (1960) [36]. Furthermore, it is possible to estimate the in vivo digestibility of the organic substance, starting from the in vitro degradability (dOM) (R2 = 0.8933, RSD = 2.7, *p* < 0.01) and from the B parameter (R2 = 0.6572, RSD = 5, *p* < 0.01).

## 4. Discussion

### 4.1. In Vivo Digestibility

The in vivo tests have provided coefficients of apparent digestibility levels of the organic matter comparable to those reported by Miraglia et al. [37], except for Diet 2. In the latter case, the low values observed were probably due to the high incidence of straw (30.6%) on the total ration. Particularly interesting are the results obtained for the digestibility of the crude fiber. For this parameter, in fact, the lowest values were recorded with Diets 4 and 5 (32.2 ± 7.32 % and 39.3 ± 4.20 %, respectively) characterized by lower forage/concentrate ratios compared to the other diets. The phenomenon could be ascribed to the greater presence of concentrates; in fact, although (according to some authors [38]) the digestibility of starch in the horse’s small intestine is equal to 85%, in contrast, Kienzle et al. [39] reported that part of the starch in cereal grains escapes pre-caecal digestion, causing an intense multiplication of amylolytic bacteria in the cecum with a consequent lowering of pH and reduction in the cellulolytic bacteria activity. As reported by Raspa et al. [40], not all the starch present in a high-cereal grain diet could be used as a source of energy; in fact, if the starch is high in the diet, it could exceed the digestive capacity of the horse intestine, and it can cause a high glycemic response. Depending on where starch is hydrolyzed in the GIT, starch is transformed into glucose by the host enzymes in the pre-caecal compartments, whereas it is degraded by microbial activity into volatile fatty acids (VFA) and lactate in hindgut fermentation chambers. The effect of the retention time on pre-caecal starch digestibility is controversial; while McLean et al. [41] and McLean et al. [42] reported an improvement in digestibility with longer retention time, in contrast, de Fombelle et al. [43] found no interaction between time and digestibility. Martin-Rosset et al. [44] also reached similar conclusions. The obtaining of the ADC of the raw fiber was superimposable to ours. The result of Diet 5 (higher digestibility of crude fiber compared to a higher content of crude fiber compared to Diet 4) certainly contributed to the higher fiber intake from the grains.

### 4.2. In Vitro Fermentation

The in vitro gas production technique could be useful in horses like ruminants for studying the nutritive value and the fermentation characteristics of diets using a cecal of feces as inoculum [44]. In the research performed by Agazzi et al. [45], the average mean retention time for feed passing through the gut of the horse was considered to be up to 38 h; however, in the present study, the incubations were extended up to 120 h. The CP content in all the diets was found to be quite low but sufficient to meet the maintenance requirement (2.8 g DNS/kg MW) [1]. In fact, even for diets with a lower protein content, while admitting a very low digestibility of 40%, we obtained a quantity of MADC equal to 315 g, which covers the requirement equal to 296 g. The same was demonstrated for the energy requirement.

As far as the in vitro tests are concerned, Diet 2, confirming the in vivo result, showed the lowest dOM, which did not correspond to a lower gas production, probably due to the high incidence of straw whose structural carbohydrates favored the activity of cellulolytics and therefore the production of gas. This hypothesis is confirmed by the intermediate value of the acetate/propionate ratio (A/P) recorded for Diet 2. Forage quality also affects digestion patterns. Immature forages (i.e., first-cut) have higher DE content and digestibility compared with later cuttings and are preferred for hard-working horses with high energy needs. On the other hand, the exclusive feeding of highly lignified fiber sources (such as straw) may increase risk of impaction colic due to the low degradation rate in the large intestine [46]. It is well known that the composition of VFA and the A/P ratio influences gas production [47]. The effects of different forage/concentrate ration (hay vs. barley) has been investigated by Julliand et al. [48]. These authors evaluated the effect of three diets (100% hay; 70% hay and 50% hay) on microbial profiles and activity, and they reported a significant decrease, both in caecal pH (6.7–6.3) and the [(acetic + butyric)/propionic] ratio, reflecting protein fermentation (4.2–3.5) and an increase in total VFA (85.2–93.0 mmol/L). Similar considerations can be made for Diet 5 that, compared with a dOM equal to 61.4%, presented a rather low value of OMCV (179 mL/g).

The high cereal content contributed to this result, which supported the development of amylolytic to the detriment of cellulolytic bacteria with consequent lower gas production. However, the degraded organic substances are partially fermented with the production of gas and VFA, but they are also used for the synthesis of microbial matter. Therefore, the dOM and the gas and VFA production of different substrates are always difficult to be compared. The results of VFA are difficult to explained. Our results contrast with Philippeau et al. [49] and Jansson et al. [50], which showed greater acetate concentrations as compared to propionate. The highest production recorded for Diet 3 and the lowest one of Diet 5 seem not influenced by the diet’s ingredients.

Lastly, the low OMCV value registered for Diet 1, which had a high forage/concentrate ratio, is difficult to interpret. However, the higher gas production recorded for diets rich in concentrates compared to forages reveals that the higher content of highly fermentable constituents in concentrates are rapidly fermented. For four diets out of five (2, 3, 4, 5), the trends of the curves (Figure 1) of the gas production as a function of time were very similar, showing a good activity of the fecal micro population during the first hours of incubation. The different trend of the gas production curve relating to Diet 1 can be attributed, at least in part, to the lower content of cereals for which the microorganisms took more time to develop and give rise to an adequate fermentation process with the relative production of gas. These differences in the degradation rate may be attributable to the chemical composition of these ingredients and to the high NDF content that ferments more slowly than starch [51].

### 4.3. Correlations between Vivo and Vitro

In agreement with Macheboeuf et al. [52], the study of the correlations made it possible to obtain regression equations to estimate the in vivo digestibility from the in vitro degradability and the kinetic parameter B. This last equation, although turning out to be significant, showed little practical value due to the high value of RSD. On the other hand, no relationship was highlighted between the in vivo digestibility and the gas produced at pre-established times. The RSDs we obtained were higher than those reported by other authors in tests in which in vivo data were compared with those obtained using the NIRS method, the chemical composition and pepsin-cellulase [38], probably also due to the small number of diets tested.

However, the important correlation between gas and VFA production highlighted in this test should be underlined. Indeed, although our study was limited to a small number of diets, in our study, we found a correlation for mmols of VFA and gas production similar to the data reported by Wolin [36] in ruminants. This author proposes a calculation to estimate the amount of gas produced considering the VFA in rumen fluid. In particular, each mmol of VFA corresponds to 1.69 mmols of gas, suggesting that the processes linked to the micro population deriving from the horse’s caecum follow metabolic pathways whose products can be modeled in the same way as for the rumen. GPT, therefore, as observed by other authors [53], represents a correct method for studying the nutritional characteristics of feed for horses, using feces as the source of inoculum, even if further investigations must be performed to improve the technique, such as lowering the RSD of the in vivo digestibility estimation equations.

## 5. Conclusions

In horses, concentrates are considered more as a hay supplement rather than a basic feed in the diet. Thus, the forage:concentrate ratio of the diet may be critical only according to the animal attitude and workload. The determination of in vivo digestibility and in vitro degradability are a common practice in ruminants to assess the nutritional characteristics of the diet, while few studies have been performed in horses, mainly regarding the in vitro gas production. This study can represent a starting point for the use of such an approach to determine and optimize a diet’s energy content in horses. Further study should focus on the evaluation of diets characterized by different forage:concentrate ratios and based of non-conventional feedstuff in order to improve a horse’s performance.

## Figures and Tables

**Figure 1 animals-13-02589-f001:**
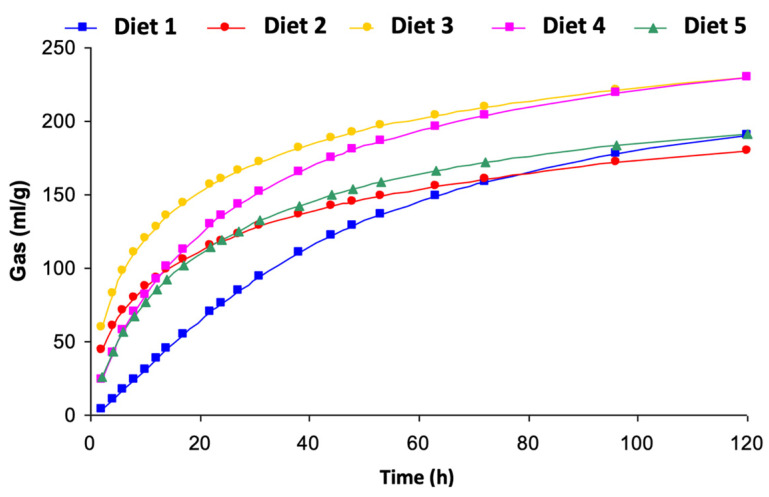
Gas production over time. Diet 1: (F/C) 90/10, Diet 2: (F/C) 78/22, Diet 3: (F/C) 68/32, Diet 4: (F/C) 60/40, Diet 5: (F/C) 50/50.

**Table 1 animals-13-02589-t001:** Ingredients and chemical composition of the five diets.

	Diet 1	Diet 2	Diet 3	Diet 4	Diet 5
	*Ingredients, %*				
Polyphite hay	90.0	47.6	47.6	47.1	35.0
Straw	-	30.6	20.5	12.6	15.0
Oat	10.0	14.6	21.4	27.2	33.4
Barley	-	7.20	10.5	13.1	16.6
	*Chemical composition*				
DM, %	89.02	86.0	86.9	89.4	85.6
Ash, % DM	10.8	7.5	6.1	6.7	6.0
CP, % DM	6.4	6.3	7.8	8.7	8.0
CF, % DM	30.0	30.7	27.1	28.3	23.6

Diet 1: (F/C) 90/10, Diet 2: (F/C) 78/22, Diet 3: (F/C) 68/32, Diet 4: (F/C) 60/40, Dieta5: (F/C) 50/50; DM: dry matter, CP: crude protein, CF: crude fiber.

**Table 2 animals-13-02589-t002:** Dry matter intake (M ± SD.) and nutritional value of the five diets.

	Diet 1	Diet 2	Diet 3	Diet 4	Diet 5
DM intake (kg/d)	11.1 ± 0.5	10.7 ± 0.4	10.9 ± 0.5	11.2 ± 0.5	10.7 ± 0.4
DM intake (g/kg MW)	105 ± 4.7	101 ± 3.8	103 ± 4.7	106 ± 4.7	101 ± 3.8
LN	1.72	1.36	1.98	1.89	1.88
Poliphyte hay (kg DM/d)	9.98 ± 0.41	5.05 ± 0.25	5.16 ± 0.20	5.25 ± 0.25	3.72 ± 0.12
Straw (kg DM/d)	-	3.32 ± 0.16	2.27 ± 0.07	1.44 ± 0.17	1.63 ± 0.06
Oat grain (kg DM/d)	1.12 ± 0.06	1.50 ± 0.04	2.33 ± 0.13	3.07 ± 0.17	3.59 ± 0.16
Barley grain (kg DM/d)	-	0.75 ± 0.03	1.12 ± 0.07	1.44 ± 0.06	1.75 ± 0.08

Diet 1: (F/C) 90/10, Diet 2: (F/C) 78/22, Diet 3: (F/C) 68/32, Diet 4: (F/C) 60/40, Diet 5: (F/C) 50/50. MW: metabolic weight. LN: maintenance nutritive level (32 g DOM/MW) [34].

**Table 3 animals-13-02589-t003:** Apparent digestibility coefficients (M± SD) of organic matter and crude fiber.

	OM	CF
	%	
Diet 1	58.8 ^A^ ± 2.6	57.2 ^A^ ± 4.2
Diet 2	46.7 ^B^ ± 2.8	44.1 ^B^ ± 2.5
Diet 3	65.5 ^A^ ± 0.6	64.8 ^A^ ± 7.6
Diet 4	61.4 ^A^ ± 3.4	32.2 ^C^ ± 7.3
Diet 5	63.4 ^A^ ± 3.7	39.3 ^BC^ ± 4.2

Diet 1: (F/C) 90/10, Diet 2: (F/C) 78/22, Diet 3: (F/C) 68/32, Diet 4: (F/C) 60/40, Diet 5: (F/C) 50/50. OM: organic matter; CF: crude fiber. A, B, C: different letters means *p* < 0.01.

**Table 4 animals-13-02589-t004:** In vitro fermentation characteristics of the five diets.

	OMCV	A	B	Yield	dOM	pH
	*mL/g*		*h*	*mL/g*	*%*	
Diet 1	175 ^B^	240 ^B^	43.0 ^A^	273 ^B^	60.0 ^AB^	6.58 ^B^
Diet 2	208 ^B^	282 ^AB^	42.8 ^A^	380 ^AB^	54.9 ^B^	6.54 ^B^
Diet 3	274 ^A^	313 ^A^	22.1 ^B^	414 ^A^	66.0 ^A^	7.18 ^A^
Diet 4	235 ^AB^	290 ^AB^	27.8 ^AB^	379 ^AB^	62.1 ^A^	6.21 ^C^
Diet 5	179 ^B^	244 ^B^	25.4 ^AB^	291 ^B^	61.4 ^AB^	6.47 ^B^
SEM	1942	1885	144	4189	21.5	0.007

Diet 1: (F/C) 90/10, Diet 2: (F/C) 78/22, Diet 3: (F/C) 68/32, Diet 4: (F/C) 60/40, Diet 5: (F/C) 50/50. OMCV: the cumulative gas production related to the incubated organic matter; A: the potential gas production; B: the time at which A/2 was formed; Yield: the cumulative gas production related to the degraded OM; dOM: organic matter degradability. A, B, C: different letters mean *p* < 0.01. SEM: medium standard error.

**Table 5 animals-13-02589-t005:** Volatile fatty acid production (M ± SD) for the five diets.

	Acetate	Propionate	Butyrate	Total VFA	A/P	(A + B)/P
	mmol/g					
Diet 1	25.3 ^CB^ ± 5.85	10.8 ^A^ ± 2.25	0.56 ^B^ ± 0.18	36.7 ^C^ ± 7.66	2.36 ± 0.49	2.41 ^B^ ± 0.50
Diet 2	27.4 ^CB^ ± 6.05	11.0 ^A^ ± 1.84	1.01 ^A^ ± 0.37	39.4 ^BC^ ± 4.48	2.62 ± 0.96	2.71 ± 0.96
Diet 3	42.0 ^A^ ± 10.7	13.1 ^A^ ± 4.00	1.37 ^A^ ± 0.56	56.5 ^A^ ± 14.5	3.26 ± 0.51	3.36 ^A^ ± 0.53
Diet 4	29.5 ^B^ ± 6.75	10.2 ^A^ ± 2.87	1.13 ^A^ ± 0.25	40.8 ^B^ ± 8.48	3.04 ± 0.94	3.16 ^A^ ± 0.95
Diet 5	21.0 ^C^ ± 7.51	7.69 ^B^ ± 1.79	0.88 ^B^ ± 0.35	29.6 ^C^ ± 8.79	2.72 ± 0.73	2.85 ^A^ ± 0.73

Diet 1: (F/C) 90/10, Diet 2: (F/C) 78/22, Diet 3: (F/C) 68/32, Diet 4: (F/C) 60/40, Diet 5: (F/C) 50/50. VFA: volatile fatty acids; A/P: acetate-propionate ratio; (A + B)/P: (acetate + butyrate)/propionate ratio. A, B, C: different letters mean *p <* 0.01.

**Table 6 animals-13-02589-t006:** Estimation equations of in vivo organic matter digestibility from in vitro fermentation.

	Eq. N.	IntercePT	b	R^2^	RSD
y = ADC (%)	1	30.504	0.5135 dOM (%)	0.8933	2.70
y = ADC (%)	2	96.824	−1.092 B (h)	0.6572	5.00
y = OMCV (mmoli)	3	2.27	1.69 AGV (mmoli)	0.8470	0.729

ADC: apparent organic matter digestibility; dOM: organic matter degradability; B: the time at which A/2 was formed; OMCV: the cumulative gas production related to the incubated organic matter; VFA: volatile fatty acids.

## Data Availability

Not applicable.

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
