# Peer review of "Forage:Concentrate Ratio Effects on In Vivo Digestibility and In Vitro Degradability of Horse’s Diet"

_animals, 2023, doi:10.3390/ani13162589_

Round 1
Reviewer 1 Report
In my opinion, it is suitable to be published in Animals. However, manuscript also need minor revision before publication.
This study evaluated the in vivo and in vitro degradability of five diets with different forage to concentrate ratios (F:C) in horses. The in vitro degradability was determined using the Gas Production Technique (GPT) with feces from the same horses used for the in vivo test as the inoculum source. The results showed no significant differences for in vivo digestibility upon different F:C content. In vitro fermentation showed a similar trend of gas production for the four out of five diets, with only an opposite trend for Diet 1, likely due to its lower cereal content. No relationship was found between in vivo digestibility and gas produced at pre-established times, but there was a correlation between gas and Volatile Fatty Acid (VFA). In my opinion, it is suitable to be published in Animals. However, manuscript also need minor revision before publication.
1. It is well known that the proportion of forage to concentrate in horse’s diet has a strong correlation with the animal’s behavior and amount of work. Table 1 should be removed. This experimental design does not include horses of varying ages or workloads;
2. In the experimental design, the nutrient content tables for the four groups of horses are described prior to the start of the experiment. The amounts of metabolizable energy, crude protein, and crude fiber under regular feeding conditions are calculated. Animals and diets determine that the horse is healthy by clarifying the horse's diet before the trial. If the diet conditions are the same as one of the 5 experimental treatments, explain this clearly in the Material method;
3. What is the data analysis method in Tables 5 and 6? It should be clearly described below in the table that the analysis should be carried out using repeated measures in the general linear model;
4. Line 238-239 sentence is incomplete. Note the error in the substance name in the fourth row of Table 3;
5. The font format of "P value" in the full text should be changed to italic;
6. Uniform the format of references;
7. The abstract should not be paragraphed and should be condensed to 200-250 characters.
The author of the article has a good English level, and there are no serious grammatical errors in the content, and only a few formatting problems.
Author Response
Please find attached the responses.

Reviewer 2 Report
Dear Authors,
Thank you for your work. You have conducted a detailed study and presented the findings in a comprehensive manner. Your work on in vivo digestibility and in vitro degradability is an important contribution to the understanding of horse diets. Your approach to using the practices common in ruminant studies to assess the nutritional characteristics of the diet in horses is novel and provides valuable insights.
However, there are some areas in your manuscript that could be improved further, mainly in updating and including more recent and relevant references. Particularly, it would be beneficial to cite recent works that have been conducted in this field, especially the research carried out in Italy, as Italy has made significant contributions to this field of study.
Including these references will not only strengthen your arguments but also provide a broader context for your study. This may necessitate revisiting your literature review process to ensure that you are aware of the latest developments in your area of study, particularly those that have been done on the diet and digestion of horses. I have adressed some specific comments throughout the manuscript.

Author Response
Thank you for improve the quality of our manuscript with your suggestions. Please find attached the pdf with all the responses.
